# Optimal Insertion Depth for Nasal Mid-Turbinate and Nasopharyngeal Swabs

**DOI:** 10.3390/diagnostics11071257

**Published:** 2021-07-14

**Authors:** Rasmus Eið Callesen, Cecilie Mullerup Kiel, Lisette Hvid Hovgaard, Kathrine Kronberg Jakobsen, Michael Papesch, Christian von Buchwald, Tobias Todsen

**Affiliations:** 1Department of Otorhinolaryngology, Head and Neck Surgery and Audiology, Rigshospitalet, Copenhagen University Hospital, 2100 Copenhagen, Denmark; recallesen@gmail.com (R.E.C.); kathrine.kronberg.jakobsen@regionh.dk (K.K.J.); christian.von.buchwald@regionh.dk (C.v.B.); 2Department of Otorhinolaryngology and Maxillofacial Surgery, Zealand University Hospital, 4600 Køge, Denmark; cecilie.m.kiel@gmail.com (C.M.K.); lisette@hovgard.dk (L.H.H.); micpa@regionsjaelland.dk (M.P.); 3Department of Clinical Medicine, University of Copenhagen, 2100 Copenhagen, Denmark; 4Copenhagen Academy for Medical Education and Simulation, University of Copenhagen and The Capital Region of Denmark, 2100 Copenhagen, Denmark

**Keywords:** SARS-CoV-2, COVID-19, antigen test, clinical validation, upper respiratory virus, virus diagnostics, mid-turbinate sample, nasopharyngeal sample, nasal swab

## Abstract

Millions of people are tested for COVID-19 daily during the pandemic, and a lack of evidence to guide optimal nasal swab testing can increase the risk of false-negative test results. This study aimed to determine the optimal insertion depth for nasal mid-turbinate and nasopharyngeal swabs. The measurements were made with a flexible endoscope during the collection of clinical specimens with a nasopharyngeal swab at a public COVID-19 test center in Copenhagen, Denmark. Participants were volunteer adults undergoing a nasopharyngeal SARS-CoV-2 rapid antigen test. All 109 participants (100%) completed the endoscopic measurements; 52 (48%) women; 103 (94%) white; mean age 34.39 (SD, 13.2) years; and mean height 176.7 (SD, 9.29) cm. The mean swab length to the posterior nasopharyngeal wall was 9.40 (SD, 0.64) cm. The mean endoscopic distance to the anterior and posterior end of the inferior turbinate was 1.95 (SD, 0.61) cm and 6.39 (SD, 0.62) cm, respectively. The mean depth to nasal mid-turbinate was calculated as 4.17 (SD, 0.48) cm. The optimal depths of insertion for nasal mid-turbinate swabs are underestimated in current guidelines compared with our findings. This study provides clinical evidence to guide the performance of anatomically correct nasal and nasopharyngeal swab specimen collection for virus testing.

## 1. Introduction

Testing is essential for controlling and limiting the spread of coronavirus disease 2019 (COVID-19), and millions of people are tested daily during the pandemic [1]. The Center for Disease Control and Prevention (CDC) recommends collecting an upper respiratory specimen for initial SARS-CoV-2 infection testing, and nasopharyngeal swabs are considered the highest-yield sample for respiratory viruses [2,3]. The clinical specimens obtained are typically tested with either a real-time reverse-transcription polymerase chain reaction (RT-PCR) or the “rapid” antigen tests for SARS-CoV-2 infection [4]. There is considerable disagreement in guidelines with respect to the recommended insertion depth of nasopharyngeal swabs [5]. The suggested insertion depth ranges anywhere from 4 to 10 cm [5,6,7,8,9]. The CDC does not describe the insertion depth for nasopharyngeal swabs, but instead states that the swab should be inserted until resistance is encountered equivalent to the posterior nasopharyngeal wall [10]. However, resistance can also be encountered before the swab reaches the posterior nasopharyngeal wall owing to a narrowed space, such as a deviated septum, or an insertion angle excessively upwards, where the swab will be blocked by the roof of the nasal cavity. Therefore, it is important to provide some guidance for the expected minimum insertion depth, so the healthcare worker knows when to expect resistance from the posterior nasopharyngeal wall during testing.

Nasopharyngeal swabs can be challenging to perform, and mid-turbinate swabs and anterior nasal swabs are increasingly used as a minimally invasive alternative for testing [11]. The main difference between the two sample techniques is the depth to which the swabs are inserted. Anterior nasal swab and mid-turbinate swab are both alternative and less-invasive techniques compared with the nasopharyngeal swab. The difference between the two is that the anterior nasal swab is only inserted 1–1.5 cm to collect material from the nasal wall, while the mid-turbinate swab is inserted around 2 cm to sample the inferior turbinate [10]. In systematic reviews, the anterior nasal and mid-turbinate swab are often grouped together as “nasal swabs”, with a sensitivity ranging from 77 to 93% compared with nasopharyngeal swabs [3]. However, the insertion depth needed to reach the mid-turbinate (defined as the middle of the inferior turbinate) is not evidence-based and knowledge about the expected insertion depth to reach the turbinate is needed.

Correct sample technique is crucial to ensure that specimen collecting is representative and to avoid false-negative test results [6,12]. A high rate of false-negative test results has been found in several studies [13]. Proper specimen collection is the first and most crucial step in COVID-19 testing [6] and correct sample technique will significantly improve the diagnostic accuracy [14]. Knowledge about the mean insertion depth to the mid-turbinate and the posterior nasopharyngeal wall is essential to correctly guide performance of an optimal collection of specimens for COVID-19 or other upper respiratory tract infection testing.

Therefore, this applied anatomy study aimed to explore the swab insertion depth required to reach the nasal mid-turbinate and posterior nasopharyngeal wall during the collection of an upper respiratory specimen for COVID-19 testing.

## 2. Materials and Methods

We conducted a prospective clinical trial to explore the swab insertion depth needed to reach the nasal mid-turbinate and the posterior nasopharyngeal wall on individuals undergoing COVID-19 testing at an urban public test center in Copenhagen, Denmark (Parken, Copenhagen Medical) on 7–8 March 2021. Individuals arriving at the test center were invited to participate in the study if they were 16 years of age or older and had no significant nose pathology. It was voluntary to participate, and both verbal and written information was provided to all individuals before they signed informed consent. The participants completed a questionnaire (registering sex, ethnicity, height, and age) prior to the nasopharyngeal swab performed for rapid antigen testing (STANDARD Q COVID-19 Ag, SD Biosensor, Gyeonggi-do, Korea). The participant was placed in a chair during the procedure, where two residents in Otolaryngology performed the test and endoscopic measurements (Figure 1B). The visualization and measurements were done with a 2.6 mm thin flexible video endoscope (ENF-V3) attached to an endoscopy processor (CV-170 with screen), both from Olympus, Tokyo, Japan. The endoscope was inserted until the posterior nasopharyngeal wall was visualized on the screen behind the participant. Then, the nasopharyngeal swab was inserted into the opposite nostril until the endoscope could visualize the tip of the swab touching the posterior nasopharyngeal wall. Here, the specimen collection was performed, and the swab length was measured from the vestibulum nasi, by holding the swab next to the vestibulum nasi and marking it with a piece of tape when retracting. The swab was then immediately measured with a ruler, after which the specimen collection from the swab was used for the “rapid” antigen tests for SARS-CoV-2 infection. Meanwhile, the endoscope was used to confirm the insertion depth to the posterior nasopharyngeal wall by placing a piece of tape on the endoscope at the tip of the nose. (Figure 1A). The endoscope was then retracted while measurements were made in the same fashion at the posterior and anterior part of the inferior turbinate and the vestibulum nasi on the way out. Finally, the distance from the vestibulum nasi to the tip of the nose was measured by marking the distance with a finger around the endoscope at the nose tip, which was immediately measured with a surgical steel ruler. The insertion depths marked with tape on the endoscope were then measured after the endoscope was retracted from the nose using a ruler. The endoscopic measurements were performed by the same registrar in otolaryngology (R.E.C.) competent in flexible pharyngo-laryngoscopy (performed >100), and the method for anatomical nasal measurements was approved and supervised by a professor (C.v.B.) and a consultant (M.P.) in rhinology. The endoscope was cleaned with a three-part decontamination system (Tristel Trio Wipes System, Tristel, Fordham, England) and proper infection prevention and control precautions were performed between every participant.

The tip-to-vestibulum length was subtracted from the endoscopic measurements of the anterior and posterior part of the inferior turbinate so all the nasal insertion depth measurements began at the vestibulum nasi. The insertion depth to the nasal mid-turbinate was calculated by adding the insertion depth to the anterior and posterior part of the inferior turbinate and dividing the sum by two. The statistical analysis was conducted using a statistical software package (R statistics version 3.6.1 [15]). Data are expressed as mean and with standard deviation (SD) or median and interquartile range, as appropriate.

## 3. Results

One hundred and nine participants were included and all 109 (100%) participants completed the endoscopic measurements. The participants’ mean age was 34.39 (SD, 13.2) years and the mean height was 176.7 (SD, 9.29) cm. Fifty-two (48%) of the participants were female (one participant refused to categorize their gender). One hundred and three of the participants (94%) were white in regards to ethnicity.

The mean insertion depth to the posterior nasopharyngeal wall and the mid-turbinate was 9.40 (SD, 0.64) cm and 4.17 (SD, 0.48) cm, respectively (see Table 1). Figure 2 shows the depths of insertion from the vestibulum nasi to the nasopharyngeal wall. All three bar charts are normally distributed.

## 4. Discussion

This study is the first to endoscopically measure the swab insertion depth for anatomically correct nasal mid-turbinate and nasopharyngeal specimen collection for COVID-19 testing.

The 9.40 (SD, 0.64) cm mean insertion depth to the posterior nasopharyngeal wall in our study is in accordance with an earlier study measuring the insertion depth for a nasopharyngeal temperature probe [16]. Previously published guidelines described a similar insertion depth between 9 and 10 cm [6], while several instruction-for-use manuals underestimate the depth down to as little as 4 cm [5,7,8,9].

In our study, the insertion depth to the posterior nasopharyngeal wall ranged from 8.0 to 10.8 cm. Some guidelines recommend using the length from the nostril to the ear to guide the insertion depth [7,8,9,17,18]. However, a previous study could not confirm such a correlation [16]. The shortest depth of insertion to the posterior nasopharyngeal was 8 cm. According to our results, the swab should be inserted at least 8 cm to reach the posterior nasopharyngeal wall in adults. It should also be noted that, in general, men had slightly longer insertion depths to all landmarks, which can also be seen in Table 1.

Overall, nasopharyngeal swabs are considered as a safe procedure, and complications are very rare. A study on more than 600,000 patients showed a complication rate of 1.24 per 100,000 performed SARS-CoV-2 tests, with the patient needing acute treatment in an emergency department [19]. Nose bleeding is the most common complication and should be kept in mind, especially if the swab meets resistance before 8 cm.

More surprisingly, we found that the mean insertion depth to mid-turbinate was 4.17 cm (SD, 0.48). These findings are also in accordance with a study on cadavers [20], but double the depth recommended for nasal mid-turbinate by CDC at about 2 cm [10]. For 39% of participants, the anterior turbinate was found at ≥2.1 cm. If we tested our population following the CDC guidelines for mid-turbinate testing, we would not make contact with the inferior turbinate in 39% of the cases. The result of this would be an anterior nasal swab test and not a mid-turbinate test. Therefore, we would recommend changing the CDC guidelines to insert the swab 4 cm (or until resistance is encountered) to ensure a mid-turbinate specimen was collected. However, we did not explore the difference in diagnostic accuracy between mid-turbinate and anterior nasal specimen collection in this study, and it should thus be explored in future studies.

Many nasal swabs have a marked line, functioning as a place to break off the sample. This line should not be used as guidance as it changes between the different brands [4]. Most swabs also have a thin part that goes beneath the inferior turbinate and a thicker part that works as a handle. The mean length from the anterior inferior turbinate to the posterior inferior turbinate was 7.40 cm in our study, indicating that the thin part of the swab should be around that length.

Self-testing is a possibility in many quick test centers. In this setting, it is even more crucial that the guidance of these individuals to test themselves is as specific and correct as possible to avoid false-negative test results. Studies searching for optimal distances as well as other anatomical papers [6] debunking myths and showing correct test techniques have great value in such a setting.

This study explores optimal techniques for collecting and testing an upper respiratory specimen for COVID-19. However, it is important to remember to repeat after a negative test if the COVID-19-like symptoms persist as the result can be false-negative. If the patient has developed severe disease, it is also important to test lower respiratory tract specimens as laryngotracheal aspiration or bronchoalveolar lavage [21,22].

Several limitations influence the interpretations of the results in our study. The insertion depth to the posterior nasopharyngeal wall was measured with the nasal swab, while the inferior turbinate measurements were performed with the flexible endoscope. Measurements with the endoscope were from the nose tip, where the tip’s length was measured afterward and subtracted. This added an extra measure, increasing the risk of a measuring error. Further, most of the participants were white, which may compromise the external generalizability of our results. However, a study with primarily Asian participants found similar nasopharyngeal measurements [16], which indicates that our results may also represent other ethnicities.

A great strength of this study is that we used an endoscope during the swab procedure to ensure correct clinical anatomical measurements. This improves the validity of our results, which can be used for future guidelines, not only for COVID-19 testing, but also for other respiratory viruses.

## 5. Conclusions

The mean optimal insertion depth was 9.40 cm (SD, 0.64) for nasopharyngeal swabs and 4.17 cm (SD, 0.48) for nasal mid-turbinate swabs. Compared with our findings, this depth is underestimated in guidelines. This study provides clinical evidence to improve the guidance of mid-turbinate and nasopharyngeal swabs for COVID-19 testing.

## Figures and Tables

**Figure 1 diagnostics-11-01257-f001:**
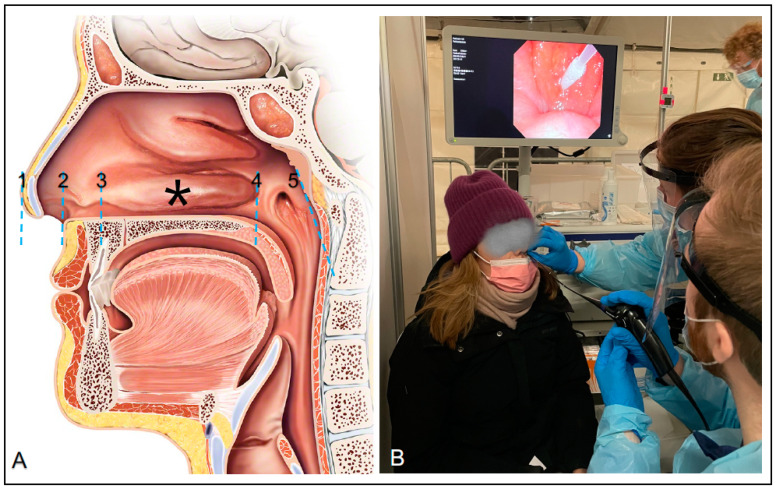
Anatomical landmarks and clinical setup. (**A**) Anatomical landmarks: 1: Tip of the nose. 2: Nasal vestibulum. 3: Anterior part of inferior turbinate. *: The calculated mid-turbinate insertion depth 4: Posterior part of inferior turbinate. 5: Posterior nasopharyngeal wall. (**B**) Picture from the clinical study setup with the endoscopically measured swab insertion depth to the posterior nasopharyngeal wall.

**Figure 2 diagnostics-11-01257-f002:**
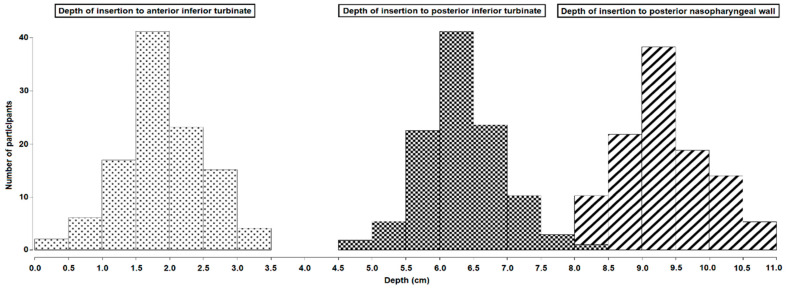
Depth of insertion to nasal landmarks. Bar charts with the insertion depth to the anterior inferior turbinate (dotted print), posterior inferior turbinate (squared print), and posterior nasopharyngeal wall (striped print). Insertion depth in cm is shown along the x-axis and number of participants along the y-axis.

**Table 1 diagnostics-11-01257-t001:** Mean insertion depths and lengths.

	Mean Insertion Depth (SD) to the Posterior Nasopharyngeal Wall	Mean Insertion Depth (SD) to the Anterior Part of the Inferior Turbinate	Mean Insertion Depth (SD) to the Posterior Part of the Inferior Turbinate	Mean Insertion Depth (SD) to the Nasal Mid-Turbinate	Mean Length (SD) from the Nose Tip to Vestibulum Nasi
All	9.40 (0.64)	1.95 (0.61)	6.39 (0.62)	4.17 (0.48)	1.42 (0.36)
Women	9.04 (0.55)	1.79 (0.47)	6.13 (0.50)	3.96 (0.39)	1.27 (0.29)
Men	9.75 (0.53)	2.09 (0.68)	6.63 (0.61)	4.36 (0.47)	1.56 (0.36)

The mean insertion depths from the vestibulum nasi to the landmarks of the nose and the mean length from the nose tip to the vestibulum nasi are seen. The insertion depth to the posterior nasopharyngeal wall is measured with the nasal swab, whereas the insertion depth to the anterior and posterior part of the inferior turbinate is measured with the endoscope, as mentioned in the methods section.

## Data Availability

Data available from the authors upon request.

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
