# Peer review of "Optimal Insertion Depth for Nasal Mid-Turbinate and Nasopharyngeal Swabs"

_diagnostics, 2021, doi:10.3390/diagnostics11071257_

Round 1

Reviewer 1 Report

The authors present "Optimal insertion depth for nasal mid-turbinate and nasopharyngeal swabs - a clinical trial", a very well written article exploring problems with the correct execution of swabs used for the detection of respiratory viruses such as SARS-CoV-2.  The manuscript is well executed and care has been taken to follow author guidelines.  I commend the authors, and have only two minor corrections, below:

  1. the use of the term "clinical trial" is misleading in this manuscript, and I suggest to the authors to remove any reference to this.  In this case the authors are making measurements for an existing and widely used technique, in the aims of improving it.  This in my opinion does not constitute a clinical trial in the modern setting, especially in a post COVID world.  Unfortunately this article does not explore the relevance of the improved technique with data from the resulting swabs - and therefore it is difficult to justify the improvements suggested in this article as improvements to the technique for detection of viruses, or reduction in false positive results.
  2. Figure 1B, I would suggest to the authors to better obscure the face of the participant

Reviewer 2 Report

intriguing study. although the pandemic is slowing down a rigth study as this one could be useful for further epidemic of other virus.

yet authors should better describe limitations/improvements regarding several points

-the high percentage during pandemic of false negative naso pharyngeal swab

-the repeated negative testing of patients with covid like syndrome (usually hospital workers)

-the advantage of this procedure compared to that suggested with bronchoalveolar lavage for highly suspected patients with a first negative swab

a fast medline to look for these few items may be helpful
